# Characteristics of Confidence and Preparedness in Paramedics in Metropolitan, Regional, and Rural Australia to Manage Mental-Health-Related Presentations: A Cross-Sectional Study

**DOI:** 10.3390/ijerph18041882

**Published:** 2021-02-15

**Authors:** Kate Emond, Melanie Bish, Michael Savic, Dan I. Lubman, Terence McCann, Karen Smith, George Mnatzaganian

**Affiliations:** 1La Trobe Rural Health School, College of Science, Health and Engineering, La Trobe University, P.O. Box 199, Bendigo 3552, Australia; m.bish@latrobe.edu.au (M.B.); G.Mnatzaganian@latrobe.edu.au (G.M.); 2Turning Point, Eastern Health, Melbourne 3120, Australia; Michael.Savic@monash.edu (M.S.); dan.lubman@monash.edu (D.I.L.); 3Monash Addiction Research Centre, Eastern Health Clinical School, Monash University, Melbourne 3128, Australia; 4Department of Nursing and Midwifery, Institute of Health and Sport, Victoria University, Melbourne 3011, Australia; terence.mccann@vu.edu.au; 5Ambulance Victoria, Melbourne 3500, Australia; karen.smith@ambulance.vic.gov.au

**Keywords:** mental health, rural, prehospital care, paramedic

## Abstract

Mental-health-related presentations account for a considerable proportion of the paramedic’s workload in prehospital care. This cross-sectional study aimed to examine the perceived confidence and preparedness of paramedics in Australian metropolitan and rural areas to manage mental-health-related presentations. Overall, 1140 paramedics were surveyed. Pearson chi-square and Fisher exact tests were used to compare categorical variables by sex and location of practice; continuous variables were compared using the non-parametric Mann–Whitney and Kruskal–Wallis tests. Perceived confidence and preparedness were each modelled in multivariable ordinal regressions. Female paramedics were younger with higher qualifications but were less experienced than their male counterparts. Compared to paramedics working in metropolitan regions, those working in rural and regional areas were generally older with fewer qualifications and were significantly less confident and less prepared to manage mental health presentations (*p* = 0.001). Compared to male paramedics, females were less confident (*p* = 0.003), although equally prepared (*p* = 0.1) to manage mental health presentations. These results suggest that higher qualifications from the tertiary sector may not be adequately preparing paramedics to manage mental health presentations, which signifies a disparity between education provided and workforce preparedness. Further work is required to address the education and training requirements of paramedics in regional and rural areas to increase confidence and preparedness in managing mental health presentations.

## 1. Introduction

Globally, the prevalence of mental illness is substantial, with 29.2% of adults experiencing mental illness in their lifetime [1]. In Australia, almost 45% of the population aged 16–85 years has experienced mental illness at some point in their life [2]. Between 2017 and 2018, the proportion of the Australian population receiving mental-health-related prescriptions was 17.5%, while 10.2% received Medicare-subsidised mental health services [3]. Non-fatal disease burden measures the number of years of healthy life lost due to disability. National data from 2015 reveal that mental and substance use disorders were the second largest contributor of non-fatal disease burden in Australia at 24%, behind musculoskeletal conditions at 25% [4]. Despite this relatively high prevalence, many people do not access services for treatment due to various structural barriers, such as availability of mental health services [5]. Paramedics are often the first point of access to health services, with mental-health-related presentations accounting for a considerable proportion of their workload in prehospital care [6,7,8]. Studies conducted in Australia report that between 9.5% and 20% of presentations to ambulance services are mental-health-related [6,9], with the lower percentages relating to state-wide studies and the higher percentage national studies. The increase in mental health presentations to ambulance services has received considerable critical attention, with paramedics providing services to fill gaps caused by deficiencies in mental health services [10,11]. The ongoing shortage of mental health professionals in Australia [12,13] signifies that service gaps may continue to be a critical issue for paramedics.

Recent studies investigating the role of paramedics indicate conflicting opinions about their scope of practice to manage mental health presentations [14,15]. It is thought that limited education and training have impacted paramedics’ confidence and preparedness, resulting in varying views about their role [14]. Studies have demonstrated a link between inadequate education and training and knowledge of legislative responsibilities including issues with interpreting and applying the Mental Health Act [16,17,18]. Previous work investigating paramedics’ perception of education and training highlighted that paramedics felt inadequately prepared when attending to mental-health-related cases [11,19]. Despite previous studies emphasising the importance of education and training in workforce preparedness for paramedics [20,21,22,23], research continues to highlight the inadequate mental-health-focused education and training for this discipline [10,11,21]. What is not yet fully understood is the degree to which paramedics feel confident and prepared to manage mental-health-related presentations, and the factors that impact this.

Utilising a cross-sectional study design, the aim of this study was to describe the perceived confidence and preparedness of paramedics to manage mental health presentations. The degree to which paramedics felt confident and prepared and the factors that were associated with this were examined. This study contributes to identified gaps in the literature on confidence and preparedness of paramedics to manage mental-health-related presentations.

## 2. Materials and Methods

### 2.1. Study Design and Participants

This study utilised an online survey targeting paramedics across Australia between April and November 2016. Ethics approval was sought from all six Australian states and two territories and was obtained from all except one state and one territory. The Human Research Ethics Committees of Eastern Health Victoria, South Australia Department for Health and Ageing, South Eastern Sydney Local Health District, and Flinders University South Australia approved the study. The remaining states and territories accepted the ethics clearance from their jurisdictions. Approval was also obtained from each ambulance service research committee or structure. All paramedics employed across the five states and one territory were invited to participate in the study. Recruitment occurred through various means including emails, e-newsletters, and bulletins from each participating ambulance service, and through Paramedics Australasia. A participant information sheet was included at the start of the survey that outlined completion as implied consent.

### 2.2. Survey Tool

A questionnaire was developed in consultation with an expert advisory group that included paramedics, paramedic educators, and mental health experts. The questions were informed by the literature and clinical practice guidelines for ambulance services. To ensure rigor, the questionnaire was finalised by consensus based on agreement reached in the expert advisory group. The final developed questionnaire measured perceived confidence and preparedness, with each question using a four-point Likert scale (1 = Not at all, 2 = A little, 3 = Moderately, 4 = Great extent). The questions specifically related to typical presentations seen in national ambulance data [6] and ambulance services’ clinical practice guidelines. Participants were asked to rate their level of confidence and preparedness in managing various mental health presentations. The confidence sub-item questions related to (1) mental health assessment; (2) mental health emergencies; (3) agitated patient; (4) acute behavioural disturbance; (5) suicide assessment; (6) mental health act; and (7) overdose. The preparedness sub-item questions related to (1) the assessment of mental health; (2) de-escalation; and (3) ability to communicate with a person experiencing a mental health crisis. Study variables collected included: age, sex, highest level of qualification, years of employment as a paramedic, location of practice, operational role, past and ongoing education, and training in mental health.

### 2.3. Statistical Analysis

The characteristics of study participants were summarised using descriptive statistics. Sexes and location of practice were compared by study categorical variables using Pearson chi-square and Fisher exact tests, whereas the non-parametric Mann–Whitney and Kruskal–Wallis tests were used to compare continuous variables. The normal distribution of continuous variables was tested using the Shapiro–Wilk test. Confidence and preparedness were modelled separately using multivariable ordinal regressions. The models adjusted for sex, age, highest level of education, years of employment as a paramedic, location of practice, and previous and ongoing training in the assessment and management of mental health states. Statistical significance was set at a *p*-value of <0.05 (two-sided) and data analysed using Stata statistical program (version 15, StataCorp, College Station, TX, USA).

## 3. Results

Overall, 1140 paramedics participated in the study; from these, full data were available on 943 (82.7%). The 197 individuals with missing data were significantly younger than those who completed all parts of the survey (*p* = 0.001); however, no substantial group differences were observed on other characteristics including sex, qualifications, operational role, region, or years of employment.

Of the remaining 943 paramedics (66.1% men), with an overall mean age of 41.3 (SD 10.9) years (ranging from 21 years to 67 years), the majority (42.2%) provided general care, followed by management tasks (15.9%), intensive care (14.6%), and other various roles (27.3%). Overall, 447 (47.4%) practised in metropolitan areas, while 253 (26.8%) and 243 (25.8%) practised in regional and rural or remote locations, respectively. There were both sex and regional differences in the characteristics of the participating paramedics. Compared with males, female paramedics were significantly younger and had higher qualifications with more graduate degrees, but they were less experienced than males and more likely to work in general care as compared to more management and intensive care roles in males (*p* < 0.001 in all) (Table 1). Regional differences were also observed with age increasing (*p* < 0.001) and level of qualification decreasing (*p* = 0.003) comparing metropolitan to regional or rural and remote locations. Operational roles differed by region with significantly more Extended Care Paramedics and Ambulance Officers found in rural and remote locations (*p* = 0.004) (Table 1). The proportion of males and females did not substantially differ by location (*p* = 0.4).

The confidence and preparedness of the paramedics to handle different mental health states considerably varied by sex and region as shown in Table 2. Compared to males, females were less confident in managing agitated patients (*p* < 0.001), patients with behavioural disturbances (*p* < 0.001) and those suffering from drug overdose (*p* = 0.002). Females were also less prepared to perform de-escalation techniques to manage patient aggression. Regional differences in confidence and preparedness were noted with less confidence and less preparedness observed among paramedics practising in rural and remote areas (Table 2).

A positive correlation was found between higher confidence and higher preparedness scores with increased degrees of prior education and training in mental health, as shown in Figure 1. No correlations were observed with age.

In the multivariable analysis on confidence, female paramedics were 10% less likely to be confident than males (adjusted-OR = 0.90, 95% CI 0.84–0.96, *p* = 0.003), while compared with paramedics practising in metropolitan locations, those in rural and remote locations showed significant less confidence in their practice (adjusted-OR = 0.87 95% CI 0.81–0.94, *p* = 0.001). Less confidence was significantly lower among paramedics aged 50 or over. Prior education on, and ongoing training in, mental health significantly increased the paramedic’s confidence, whereas level of education and operational role were not associated with confidence (Table 3).

In the multivariable model of preparedness, no age and no sex differences were observed. However, preparedness of the paramedic was significantly lower among paramedics practising in rural and remote areas (adjusted-OR = 0.83, 95% CI 0.76–0.92, *p* < 0.001). Prior education on, and ongoing training in, mental health were more likely to increase the paramedic’s preparedness to perform the needed skills to handle patients experiencing mental health states. Level of highest qualification had no association with this outcome.

## 4. Discussion

This study reports sex and regional differences in confidence and preparedness of paramedics to manage mental-health-related presentations. Prior research on sex differences in paramedics is limited. Key findings from the 2011 census data on paramedics reported that females comprised 32% of the paramedic workforce in 2011 compared to 26% in 2006, and in the 20–29 age group, 53% of paramedics were female [24]. In a 2016–2017 report, 56.7% of new paramedic recruits were female [25]. Consistent with these reports, this study found that female paramedics were younger; however, compared to their male counterparts, they were less confident in managing mental health presentations, particularly those relating to agitated behaviours. Previous research has found significant sex differences in performance and leadership when managing challenging situations related to cardiopulmonary resuscitation (CPR), with men tending to emerge as leaders [26]. The authors recommended that future training in CPR should focus on leadership skills for females, to disband the observed differences between the sexes [26]. However, caution must be applied when considering sex differences. There has been criticism that differences between males and females may reflect gender role expectations rather than sex differences [27]. Furthermore, male roles have been associated with dominance and leadership, which may prevent females from displaying leadership behaviours [27]. Paramedicine has typically been a male-dominated profession (19, 24, 25), and females are expanding into traditionally male-dominated health professional roles [28]. A report by Mason [29] highlighted ongoing barriers to female leadership in paramedicine and emphasised the culture of gender bias needed to change. It is possible that culture and gender role expectations may be influencing females’ confidence in managing complex mental health presentations.

Of equal importance to sex differences was the finding that paramedics working in rural and regional areas were significantly less confident and prepared to manage mental-health-related presentations in comparison to metropolitan counterparts. There is a paucity of research on how paramedics manage mental health presentations in rural areas. Previous research examining the role of paramedicine in rural areas has identified a need to tailor educational requirements to maintain skills that are not often utilised [30]. An Australian study that focused on one area health service examined ambulance use in metropolitan and rural areas. A study focusing on one area health service found a disparity between rates of ambulance use with rural communities being less likely to use services [31]. Decreased exposure to mental-health-related presentations may explain why rural and regional paramedics feel less confident and prepared. This has been illustrated in previous research where decreased exposure to specific clinical presentations such as cardiac arrest has correlated with a decline in clinical skills [32]. It has been suggested that strategies to increase exposure, such as simulation training, should be explored [32]. Indeed, it has been identified that rural paramedics require education and training specific to their needs, as rural practice has a focus on community rather than the types of cases [33,34]. This community focus is further explained in a recent Australian study that describes rural adversity as impacting mental health and wellbeing, requiring a collaborative and integrated approach with local leadership and community support [35]. Paramedics in rural areas have adopted a multidisciplinary approach to health care that has extended their role beyond specific skills, to include community engagement in primary health care, supporting various health disciplines and volunteers, and participating in education and training across health disciplines, also known as interprofessional learning [30,34,36,37]. The extended role of paramedicine in rural areas is due in part to workforce recruitment and retention in rural health care [20,23,36]. Although interprofessional learning may be regarded as an ideal environment to explore the alliance between health professionals in the delivery of health care [36], there is a shortage of mental health professionals [38,39] to contribute to this, further impacting the transference of skills and knowledge for education and training in mental health. The extended role of paramedicine towards a wider community focus, coupled with decreased exposure to specific clinical presentations and a shortage of specialist mental health professionals to provide education, may consequently be impacting confidence on preparedness when managing mental-health-related presentations.

The role of paramedicine in prehospital care is complex, and has changed significantly, with speculation as to whether paramedic education has kept pace with these changes [10]. The findings from this study identified that prior education on, and ongoing training in mental health significantly increased paramedics confidence, whereas level of education and operational role were not associated with confidence. These results are consistent with previous research that has identified a correlation between focused clinical training and increased confidence levels exemplified by increased levels of confidence by paramedics with more education and training in paediatric emergencies [40]. This reinforces the importance of ongoing education and training and its impact on health professionals’ levels of confidence and preparedness.

Strengths of this study include the representation of a national sample of paramedics who completed the survey anonymously. Limitations include the self-report, cross-sectional nature of the data, potentially allowing bias in the responses. In addition, as a convenience sample, the responses may not be generalisable to the broader paramedic population.

## 5. Conclusions

This study examined confidence and preparedness of paramedics to manage mental-health-related presentations, and the results identified significant sex and regional differences. The portion of the paramedic workforce comprising younger females requires additional support to improve confidence. This support may go beyond that of providing education on mental health skills, to leadership skills and attributes, with the potential change of gender-defined roles and culture. Ongoing education and training are needed in regional, rural, and remote areas to increase confidence and preparedness, with a considered community focus to reflect specialist workforce shortages and the expanded roles identified in rural practice contexts. The findings of this study may indicate that higher qualifications from the tertiary sector are not adequately preparing paramedics to manage mental-health-related presentations, signifying a disparity between education provided and workforce preparedness. Further work is needed to align curriculum to workforce requirements. Future research must focus on specific strategies to tailor education and training for paramedics in both undergraduate curriculum and ongoing professional development to improve confidence and preparedness when managing mental-health-related presentations, particularly for the rural and regional workforce. Ongoing professional development is a priority as from 2018, paramedics are a regulated profession under the National Registration and Accreditation Scheme, setting a standard to improve competence and patient outcomes.

## Figures and Tables

**Figure 1 ijerph-18-01882-f001:**
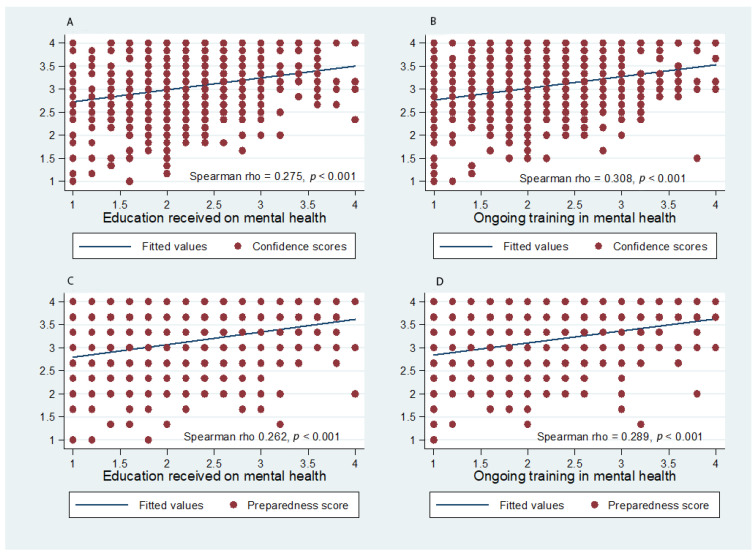
(**A**–**D**) Confidence and preparedness scores by received and ongoing education on and training in mental health.

**Table 1 ijerph-18-01882-t001:** Characteristics of paramedics by sex and region of practice.

	All	Male	Female	*p*-Value	Metropolitan	Regional	Rural/Remote	*p*-Value
	*n* = 943 (100.0%)	*n* = 623 (66.1%)	*n* = 320 (33.9%)		*n* = 447 (47.4%)	*n* = 253 (26.8%)	*n* = 253 (26.8%)	
Age, mean (SD)	41.3 (10.9)	43.8 (10.7)	36.4 (9.5)	<0.001	39.6 (10.3)	42.2 (10.8)	43.3 (11.6)	<0.001
Age categories, %								
20–29 years	18.7	13.3	29.1	<0.001	21.9	14.2	14.2	<0.001
30–39 years	24.7	20.4	33.1		27.7	24.1	24.1	
40–49 years	31.5	33.2	28.1		30.4	36.4	36.4	
50 years or older	25.1	33.1	9.7		19.9	25.3	25.3	
Highest qualification, %								
Certificate	1.2	1.1	1.3	<0.001				0.033
Diploma	35.4	41.6	23.4		0.9	1.6	1.6	
Bachelor degree	42.5	35.6	55.9		31.5	35.2	35.2	
Graduate certificate	6.2	6.1	6.2		43.6	43.1	43.1	
Graduate diploma	9.9	9.9	9.7		6.7	4.3	4.3	
Master or higher	4.2	5.0	2.8		11.2	11.1	11.1	
Missing	0.6	0.6	0.6		5.8	3.9	3.9	
Years employed as a paramedic, %								
1 to 5 years	38.4	34.3	46.2	<0.001	40.9	40.9	35.0	0.1
6 to 10 years	22.8	20.6	27.2		24.2	24.2	21.8	
11 years or more	36.6	43.5	23.1		33.1	33.1	39.1	
Missing	2.2	1.6	3.4		1.8	1.8	4.1	
Operational role, %								
Paramedic Manager	15.9	18.9	10.0	<0.001	14.3	19.8	14.8	0.004
Intensive Care Paramedic	14.6	16.7	10.6		15.9	15.8	11.1	
Extended Care Paramedic	8.1	8.0	8.1		7.6	5.1	11.9	
General Care Paramedic	42.2	38.8	48.7		43.2	39.5	43.2	
Ambulance Officer	5.2	4.8	5.9		2.9	6.7	7.8	
Other	14.0	12.7	16.6		16.1	13.0	11.1	

**Table 2 ijerph-18-01882-t002:** Confidence and preparedness by sex and region of practice (scores reported as means and standard deviations)

	Male	Female	*p* Value	Metro	Regional	Rural	*p* Value
	*N* = 623 (66.1%)	*N* = 320 (33.9%)		*N* = 447 (47.4%)	*N* = 253 (26.8%)	*N* = 243 (25.8%)	
Confidence in managing:							
Mental state assessment	3.11 (0.60)	3.11 (0.69)	0.9	3.14 (0.62)	3.11 (0.65)	3.06 (0.64)	0.3
Mental state emergencies	3.16 (0.62)	3.12 (0.70)	0.3	3.17 (0.62)	3.18 (0.68)	3.07 (0.66)	0.1
Agitated patient	3.16 (0.66)	2.91 (0.78)	<0.001	3.12 (0.69)	3.14 (0.73)	2.91 (0.71)	<0.001
Acute behavioural disturbances	3.07 (0.71)	2.88 (0.77)	<0.001	3.05 (0.73)	3.08 (0.75)	2.85 (0.73)	<0.001
Suicide assessment	3.06 (0.76)	3.11 (0.75)	0.4	3.07 (0.77)	3.13 (0.74)	3.04 (0.73)	0.4
Mental Health Act	3.00 (0.74)	2.83 (0.74)	0.007	2.96 (0.74)	2.96 (0.77)	2.81 (0.71)	0.018
Overdose	3.55 (0.56)	3.43 (0.58)	0.002	3.56 (0.56)	3.52 (0.57)	3.40 (0.56)	0.002
Preparedness to perform the following skills:							
De-escalation	3.24 (0.66)	3.04 (0.77)	<0.001	3.22 (0.70)	3.22 (0.66)	3.04 (0.75)	0.003
Communication with a person experiencing a mental health crisis	3.24 (0.65)	3.19 (0.72)	0.3	3.23 (0.65)	3.23 (0.67)	3.11 (0.70)	0.006
Mental health risk assessment	3.03 (0.72)	3.06 (0.76)	0.6	3.04 (0.75)	3.12 (0.68)	2.96 (0.74)	0.1

**Table 3 ijerph-18-01882-t003:** Factors predicting confidence and preparedness of paramedics to manage mental health states: multivariable analyses.

	Model 1: Confidence			Model 2: Preparedness		
Covariates	OR	95% CI	*p*-Value	OR	95% CI	*p*-Value
Age categories						
20–29 years (reference)	1.00			1.00		
30–39 years	1.03	0.93–1.13	0.6	1.07	0.95–1.20	0.3
40–49 years	0.99	0.89–1.09	0.9	1.03	0.91–1.17	0.6
50 years or over	0.87	0.77–0.98	0.027	0.96	0.83–1.11	0.6
Female	0.90	0.84–0.96	0.003	0.94	0.86–1.02	0.1
Location of practice						
Metropolitan (reference)	1.00			1.00		
Regional	0.98	0.91–1.06	0.6	0.97	0.89–1.06	0.5
Rural/remote	0.87	0.81–0.94	0.001	0.83	0.76–0.92	<0.001
Highest qualification						
Certificate (reference)	1.00			1.00		
Diploma	0.95	0.70–1.28	0.7	1.02	0.71–1.45	0.9
Bachelor degree	1.04	0.77–1.41	0.8	1.06	0.74–1.52	0.7
Graduate certificate	1.05	0.76–1.45	0.7	1.01	0.68–1.48	0.9
Graduate diploma	1.21	0.88–1.66	0.2	1.24	0.86–1.81	0.2
Master or higher	1.12	0.80–1.56	0.5	1.03	0.69–1.53	0.9
Years employed as a paramedic						
1 to 5 years (reference)	1.00	1.00				
6 to 10 years	1.13	1.13	0.003		1.02–1.24	0.019
11 years or more	1.05	1.05	0.3		0.87–1.07	0.5
Overall score of prior education on mental health	1.15	1.09–1.22	<0.001	1.19	1.11–1.27	<0.001
Overall score of ongoing education in mental health	1.19	1.14–1.26	<0.001	1.22	1.15–1.29	<0.001
Operational role						
Paramedic Manager (reference)	1.00			1.00		
Intensive Care Paramedic	1.03	0.92–1.15	0.6	0.98	0.86–1.13	0.9
Extended Care Paramedic	1.13	0.99–1.29	0.1	1.18	1.01–1.39	0.037
General Care Paramedic	0.97	0.88–1.07	0.5	0.92	0.82–1.04	0.2
Ambulance Officer	0.93	0.79–1.09	0.4	0.95	0.79–1.15	0.6
Other	0.90	0.80–1.01	0.1	0.89		0.1

## Data Availability

All data relevant to the study are included in the article.

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
