# Peer review of "Characteristics of Confidence and Preparedness in Paramedics in Metropolitan, Regional, and Rural Australia to Manage Mental-Health-Related Presentations: A Cross-Sectional Study"

_ijerph, 2021, doi:10.3390/ijerph18041882_

Round 1

Reviewer 1 Report

Overall Comments

It is an important topic, given the prevalence of mental illness and the barriers to seeking mental healthcare. However, I have concerns about the interpretations of the results on preparedness and think that the authors need to rule out other explanations before drawing the present conclusions. Please see below.

Introduction

-I would like more elaboration on what inadequacies have been documented on mental health education and training for paramedics (lines 58-61). I also would like more background literature that would support hypothesis of the study. What did previous studies find about confidence and preparedness of female vs. male paramedics? If there aren’t studies on this yet, what were the conflicting opinions about their role in addressing mental health?

Method

-Authors stated that the questionnaire drew on existing published scales (line 84). Which scales? Are these from the national ambulance data? What did you mean by “questions specifically related to”? Did you use the same items verbatim or modify them? If you created new questionnaire and consulted with expert advisory group, who were part of this advisory group? Are they authors of the study or are they independent of the study?

Results

-Authors reported that there were no sex or age differences for preparedness in multivariate analyses, whereas there were sex differences for confidence. Could this just be confidence level or gender role of females, independent of their actual skills?

-Did you find an interaction effect of sex and region on confidence or preparedness? Or an interaction effect of sex and prior education on confidence and preparedness?

Discussion

-I’m confused by the statements throughout manuscript (e.g., lines 25-26, 181-182) that sex differences were found for preparedness. In your multivariate model controlling for prior education and other covariates, you showed data that there weren’t sex and age differences. Instead, the preparedness seem to depend on prior training and geographic location.

-I agree that the sex differences in confidence should be interpreted with consideration of gender role expectations. Just because females are not as confident as males, it does not mean they are less competent at the job.

-It would be helpful if authors can rule out or address the following explanations:

1) Could the lower levels of confidence reported be explained by females’ gender role expectations? They may know what they are supposed to do, but they just don’t claim to know it all. In that case, it changes the implications of the study. It is not that females are unprepared. It's just they need more reassurance that they are doing a good job.

2) Would you still find main effects of sex and age if you do an interaction of prior experiences or training? It seems to me that prior experiences and training play a big role in confidence and preparedness.

Reviewer 2 Report

Dear authors, thank you very much for your submission to the IJERPH and for giving me the opportunity to review your interesting manuscript on paramedics’ confidence and preparedness. Your research is appreciated as it sheds light on this research gap. Your findings are relevant and interesting. Your paper shows good quality in all regards. However, I found several issues that should be addressed before publication in the IJERPH can be considered:

  1. The journal does not apply a structured abstract style. Therefore, rather than writing “Objective: …”, I’d recommend to write: “This study aims to…”. Similarly for Methods etc.
  2. Please also briefly mention the methods, results, and contributions of your study at the end of the introduction.
  3. Please clearly state your results in one summarizing sentence in the results and conclusion sections.
  4. Please check if references published in IJERPH might also be cited.

I hope you find my comments helpful. Good luck with your revision!

Round 2

Reviewer 1 Report

Thank you for addressing previous comments. I don't have any other concerns.